# Analysis of Flow and Wear Characteristics of Solid–Liquid Two-Phase Flow in Rotating Flow Channel

Peng Wang [ID], Xinyu Zhu and Yi Li *

State-Province Joint Engineering Lab of Fluid Transmission System Technology, Zhejiang Sci-Tech University, Hangzhou 310018, China; wangpengzstu@outlook.com (P.W.); zhuxinyuzstu@outlook.com (X.Z.)
* Correspondence: liyi@zstu.edu.cn; Tel.: +86-1364-684-6259

**Abstract:** To study the flow characteristics and the wear distribution of pumps at different rotation speeds, a rotating disc with three blades was designed for experiments. Numerical simulations were conducted using a computational fluid dynamics-discrete phase model (CFD–DPM) approach. The experimental and numerical results were compared, and the flow characteristics and wear behaviors were determined. As the speed increased, the particles at the blade working surface aggregated. The particle velocity gradually increased at the outlet of the channel. The severe wear areas were all located in the outlet area of the blade working surface, and the wear area extended toward the inlet area of the blade with increasing speed. The wear rate of the blade surface increased as the speed increased, and an area with a steady wear rate appeared at the outlet area of the blade. When the concentration was more than 8%, the severe wear areas were unchanged at the same speed. When the speed increased, the severe wear areas of the blade produced wear ripples, and the area of the ripples increased with increasing speed. The height difference between the ripples along the flow direction on the blade became larger as the speed increased.

**Keywords:** solid–liquid two-phase flow; rotating disc; large particles; wear; CFD–DPM

## 1. Introduction

Centrifugal slurry pumps are widely used for the transport of solid materials in mining, water conservation applications, power generation, dredging, the chemical industry, and other industries. These pumps have been developed by modifying the conventional centrifugal pump design to ensure the smooth delivery of solid–liquid two-phase flow [1]. Wear is an important consideration in the design of these pumps, and it is related to the service lives of the pump components. A better understanding of the wear would help to improve the pump performance and reliability. The main parts of the pump that become damaged are the impeller and the volute. Therefore, in the process of solid–liquid two-phase flow, impeller wear is inevitable, and considerable research on this subject has been carried out in recent decades.

Many scholars have studied the solid–liquid two-phase flow wear. Tarodiya et al. [2] analyzed the internal wear position and wear mechanism of a centrifugal pump. Xiao et al. [3] used numerical simulations to study the effects of geometric model changes on the flow characteristics and wear mechanism of centrifugal pumps after impeller wear caused by 0.5-mm-diameter solid particles. The impeller wear first affected the flow characteristics, and then wear occurred. Khalid et al. [4] used an experimental method to analyze the relationship between the impeller weight loss, blade height loss, blade thickness loss, and impeller diameter loss over time due to 1.12 mm diameter solid particles. Xing et al. [5] studied the wear positions in centrifugal pumps. The particle size, impact speed, impact

angle, and taper of the particles had important influences on the position and amount of wear of the impeller. Tressia et al. [6] studied the effect of different particle sizes (0.15–2.4 mm) on the wear resistance of steel in solid–liquid two-phase flow. They found a critical particle size. When the critical size was exceeded, there was a linear relationship between the particle diameter and the mass loss caused by the wear. Chandel et al. [7] used a Coriolis test bed and found that the sizes of the solid particles (4.25–300 μm) and the rotation speed had significant influences on the wear. Tian et al. [8] designed a wear test bench to use solid particles with 660-μm-diameter to determine the material wear coefficient under different erosion conditions. They effectively predicted the wear life of the mud pump over-flow components. Azimian et al. [9] designed a centrifugal accelerated erosion wear test machine. With the increase in the flow and rotation speed, the erosion showed an upward trend. Serrano et al. [10] found that wear experiments using small-scale particles were better for studying the wear properties of sediment. Changes in the particle shape led to micro-grooves and plastic deformation, and the decrease in the impeller speed led to a decrease in the wear. Deng et al. [11] conducted a detailed experimental study on the influence of solid particles on the corrosion wear using a centrifugal accelerated erosion test device. Li et al. [12] found that the wear of the solid particles on the centrifugal pump was relatively small when the particle volume fraction was less than 2.5% using experimental and simulation methods, and proposed a scheme to reduce the wear. Peng et al. [13] found a linear relationship between the concentration of the slurry and the wear of the impeller blades and guide vanes using numerical simulations. Through the research of these scholars, it was seen that the particles size and rotation speed of the impeller can significantly affect the flow characteristics and wear laws.

The flow characteristics of the two-phase flow influence the wear behavior. Cai et al. [14] found that the application of vorticity and the Q-criterion in the study of closed impellers could effectively explain the wear behavior and mechanism of the impeller. Liu et al. [15] used the Euler model and concluded that the distribution of solid particles in the impeller passage was mainly affected by the particle size. Through numerical simulations, Zhao et al. [16] found that the particles in the impeller passage were mainly concentrated on the side of the pressure surface under non-rated conditions. Pagalthivarthi and Gupta et al. [17] used discrete phase model of Fluent to determine the erosion wear in centrifugal pump casing pumping dilute slurries. The casing geometry is considered two-dimensional. They presented the wear characteristic of casing with various parameters to lower the wear rates and to make the wear pattern along the casing wall as uniform as possible.

Scholars have studied the wear behaviors caused by various factors through experimental and simulation methods. In terms of the impact of particle size, most research is carried out on small particles. However, when the particle size is large, its motion law is very different from that of small particles, especially in a rotating flow channel. Additionally, this will have a greater impact on the law of wear. So, the research on the impact of large particle was carried out in this article on the conditions of different rotating speed. In this paper, a wear test bench with an adjustable speed disc was designed to study the wear behavior of the blade surface for various disc speeds. The flow in the test unit was numerically simulated using computational fluid dynamics (CFD), and the results were compared and verified with the experimental data.

## 2. Experimental Methods

The experimental platform included a rotating disc that simulated a closed impeller. The frequency of the rotating disc was adjusted by a variable frequency motor connected to a frequency converter. A fixed flow rate of the two-phase flow was achieved by a centrifugal pump, and the fluid was sent to the rotating disc for the wear experiments. Figure 1 shows a schematic view of the experimental bench system, and Figure 2 shows a photograph of the experimental table.

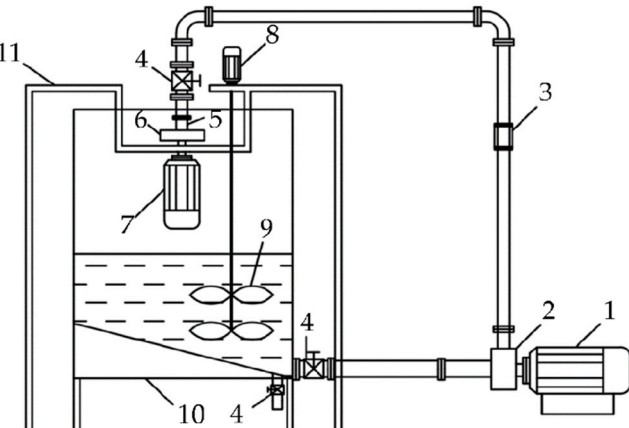

**Figure 1.** Schematic view of the experimental bench system. 1. Motor; 2. pump; 3. electromagnetic flow meter; 4. valve; 5. sliding bearing; 6. rotating disc; 7. frequency conversion motor; 8. stirring motor; 9. stirrer; 10. water storage tank; 11. motor support.

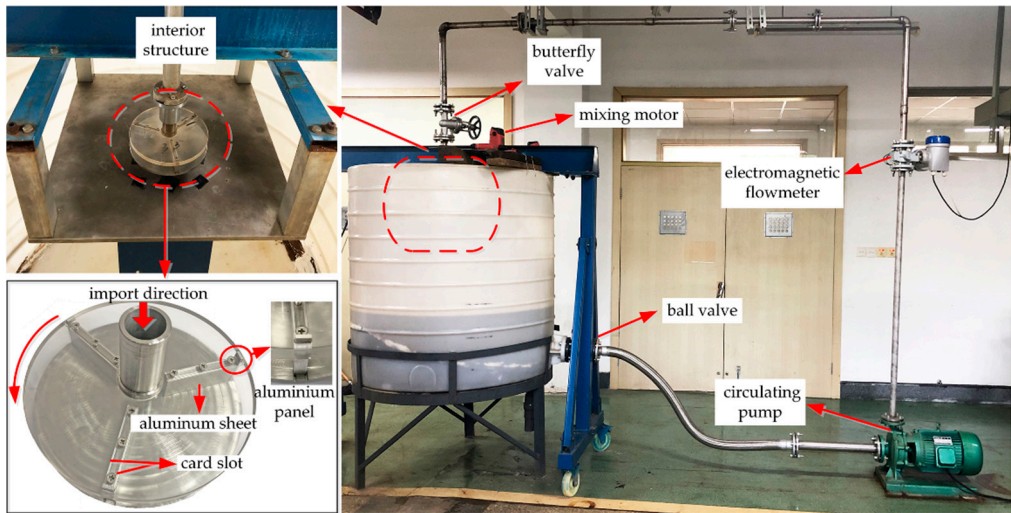

**Figure 2.** Photographic view of the experimental table.

Figure 2 shows the rotating disc with a simulated closed impeller design. The disc was composed of three blades. Due to the relatively large number of experimental conditions, to ensure that the wall was in the same initial state for each experiment, replaceable wall-attached aluminum sheets were used for the wear experiments. A blade clamping slot was used to fix the experimental aluminum sheet. After completing the test with one set of working conditions, the experimental aluminum sheet was replaced. An aluminum sheet baffle was designed at the outlet end of the blade to prevent the experimental aluminum sheet from being thrown out of the clamping slot. The surface roughness (Ra) of the aluminum sheet was given by the supplier as 3.2, and the size of the aluminum sheet was 68 mm × 23 mm. Before the experiment, six pieces of aluminum were successively marked and installed in the card slot of the disk, and clean water was added to the water tank. After calculating the particle mass to be added, the water was added to the tank. After turning on the agitator, the frequency converter was turned on to adjust the input frequency and speed of the variable frequency motor. The valve and circulating pump were opened in sequence to start the experiment.

Figure 3a shows the custom soda-lime glass used for the experiment. The particles were spherical, and the particle density was 2700 kg/m$^3$. One hundred particles were randomly selected for the diameter measurements. The particle size distribution results are shown in Figure 3b. The particle diameters were between 2.85 and 3.15 mm. The fit curve shows that the particle diameter in the sample

was normally distributed, which met the experimental requirements. The rotation speed and the mass concentration of solid particles were varied in the experiments. The two-phase mixture concentrations were 2%, 4%, 6%, 8%, and 10%, and the speeds were 400, 500, 600, 700, and 800 rpm. Twenty-five experiments were conducted, and each experiment was conducted for 3 h.

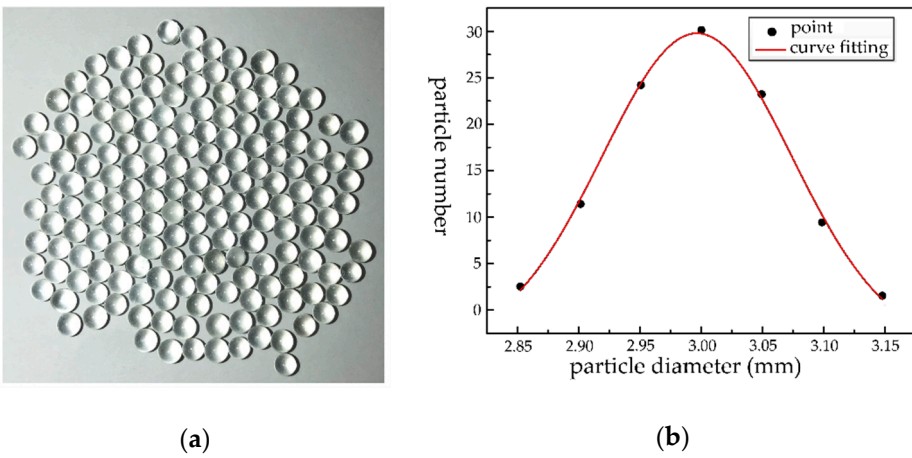

(**a**)　　　　　　　　　　　　　　　　　　　　(**b**)

**Figure 3.** (**a**) Glass particles; (**b**) particle diameters.

To determine the thickness loss of the worn aluminum sheet, a laser displacement sensor was used to measure the wear depth of the worn aluminum sheet surface. The laser displacement thickness measurement was produced by KEYENCE of Japan. The controller model is LK-G5001V, and the laser generator is LK-H020. The highest measurement accuracy of the sensor is 1 μm. The laser displacement thickness measurement system and the measurement method are shown in Figure 4. Due to the large wear area, it was difficult to measure the thickness at all positions. So, three straight lines were finally selected for the measurements. The distance between upper/lower line and the center line is 7 mm, respectively. Before the wear experiment, the thickness of the aluminum sheet at the marked point was measured. After the experiment, the thickness of the aluminum sheet at the same position was measured again. The difference between the two thickness measurements was the wear depth. In order to reduce the random error, each point was measured three times. Additionally, the average value was taken as the experimental result.

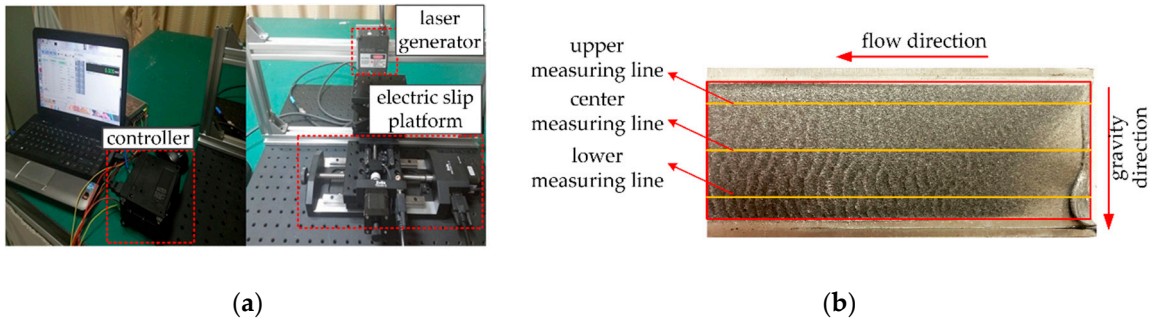

(**a**)　　　　　　　　　　　　　　　　　　　　(**b**)

**Figure 4.** (**a**) Laser displacement sensor; (**b**) measurement method.

## 3. Model Design and Numerical Method

### 3.1. Computational Domain Model

Solidworks was used to model the hydraulic model of the rotating disk to simulate the water flow. The model included an inlet extension and a rotating disk. The inlet extension was designed to ensure the full development of the solid–liquid two-phase flow and improve the calculation accuracy.

As shown in Figure 5, the inlet extension section was 28 mm in diameter and 210 mm in length. The rotating disc contained three blades of the same size. The blade was 68 mm long and 8 mm wide. The diameter of the rotating disc was 200 mm, and the height was 23 mm.

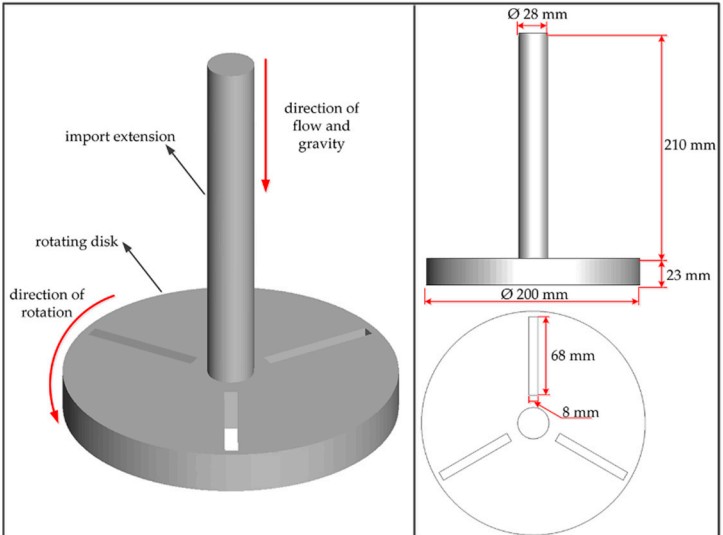

**Figure 5.** Computational domain of the centrifugal slurry pump.

*3.2. Numerical Methods*

The flow field of the solid–liquid two-phase flow was calculated based on a Eulerian–Lagrangian method, in which the equations of motion for the two phases were solved separately. In this method, the fluid phase was treated as a continuous phase and a large number of particles in the flow field were processed into discrete phases.

3.2.1. Mathematical Formulation

The governing equation of the solid–liquid two-phase is introduced.

The continuity equation expresses conservation of mass, which states that the mass change per unit time of the fluid control volume is equal to the net flux across the control surface. The specific mathematical expression is as follows:

$$\frac{\partial \rho}{\partial t} + \frac{\partial (\rho u_x)}{\partial x} + \frac{\partial (\rho u_y)}{\partial y} + \frac{\partial (\rho u_z)}{\partial z} = 0, \tag{1}$$

where $\rho$ is the density of the fluid, $t$ is the time, and $u_x$, $u_y$, and $u_z$ are the components of velocity vector of the fluid in the x-, y-, and z-directions, respectively.

The conservation of momentum equation is as follows:

$$\begin{aligned}
\rho \frac{Du_x}{Dt} &= \rho f_x - \frac{\partial p}{\partial x} + \mu \left( \frac{\partial^2 u_x}{\partial x^2} + \frac{\partial^2 u_x}{\partial y^2} + \frac{\partial^2 u_x}{\partial z^2} \right) \\
\rho \frac{Du_y}{Dt} &= \rho f_y - \frac{\partial p}{\partial y} + \mu \left( \frac{\partial^2 u_y}{\partial x^2} + \frac{\partial^2 u_y}{\partial y^2} + \frac{\partial^2 u_y}{\partial z^2} \right) \\
\rho \frac{Du_z}{Dt} &= \rho f_z - \frac{\partial p}{\partial z} + \mu \left( \frac{\partial^2 u_z}{\partial x^2} + \frac{\partial^2 u_z}{\partial y^2} + \frac{\partial^2 u_z}{\partial z^2} \right)
\end{aligned} \tag{2}$$

where $\rho$ is the density of the fluid, $t$ is the time, $u_x$, $u_y$, $u_z$ are the components of velocity, $f_x$, $f_y$, $f_z$ are components of the external force, $p$ is the eqivalent pressure and $\mu$ is the liquid-phase dynamic viscosity.

Based on Newton's second law, the motion equation of solid particles in the flow field is as follows:

$$
\begin{cases}
m_p \frac{du_x}{dt} = mg_x + F_{D_x} + F_{p_x} + F_{v_x} \\
m_p \frac{du_y}{dt} = mg_y + F_{D_y} + F_{p_y} + F_{v_y} \\
m_p \frac{du_z}{dt} = mg_z + F_{D_z} + F_{p_z} + F_{v_z}
\end{cases}
\tag{3}
$$

where $m_p$ is the particle quality, $u_x$, $u_y$, $u_z$ are the components of velocity, $g_x$, $g_y$, $g_z$ are the components of gravitational acceleration, $F_{D_x}$, $F_{D_y}$, $F_{D_z}$ are the components of drag force, $F_{p_x}$, $F_{p_y}$, $F_{p_z}$ are the components of additional quality force, and $F_{v_x}$, $F_{v_y}$, $F_{v_z}$ are the components of pressure gradient force.

The coupling between the continuous phase and the discrete phase are conducted based on DPM method. In the DPM method, physical quantities of particles, such as the change of momentum, are tracked and calculated along particle trajectory. The change of momentum is the momentum value transferred from the continuous phase to the particle discrete phase. Then these physical quantities are used in the subsequent continuous phase calculations. The governing equations of the discrete phase (Equation (3)) and the continuous phase (Equation (2)) are solved alternately until both converge (that is, the two calculation solutions do not change). So, the two-way coupling calculation is realized.

By comparing the calculation results of various wear models with experimental data, the wear model proposed by scholar Ahlert [18] was finally selected. This model is suitable for wear materials, such as aluminum alloy and carbon steel. The equations of the model are as follows:

$$
ER = A(BH)^{-0.59} F_s u_p^{n_A} f(\theta),
\tag{4}
$$

$$
f(\theta) =
\begin{cases}
a\theta^2 + b\theta & \theta \le \theta_0 \\
x \cos^2\theta \sin(w\theta) + y \sin^2\theta + z & \theta > \theta_0
\end{cases},
\tag{5}
$$

where $ER$ is the erosion value, $A$ is a constant related to the wall material, $BH$ is the Brinell hardness of the wall material, $F_s$ is a coefficient related to the shapes of the particles (spherical particles have values of 0.2, and sharp particles have values of 1), $u_p$ is the particle collision speed, and $f(\theta)$ is a function of the particle collision angle.

### 3.2.2. Simulation Parameters

For the numerical simulations in FLUENT, the parameters must be set. The Navier–Stokes (N–S) equations were solved, and the turbulence model was the renormalization group (RNG) k-$\varepsilon$ model. The CFD–DPM (computational fluid dynamics–discrete phase model) model in FLUENT was used. Solid–liquid two-phase flow and wear simulations of a rotating disk at different particle mass concentrations and speeds were carried out. The inlet and outlet boundary conditions were a velocity inlet and an outflow condition. The SIMPLEC velocity–pressure coupling method was used to solve the fluid flow equations. The transient time step was the time required for the impeller to rotate 2°, and the calculation time was 0.75 s.

### 3.3. Mesh Generation

Ansys ICEM was used to generate the grid in the computational domain of the rotating disk. The structure of the rotating disk entrance extension was simple, and the rotating disk itself was more complicated. Thus, a mixed grid was used. A structural grid was used for the extension section at the entrance of the rotating disk, while an unstructured grid was used for the rotation disk itself. The quality of the grid was greater than 0.4, and the minimum angle of the grid was 28°, which met the calculation requirements. The rotating disc mesh is shown in Figure 6.

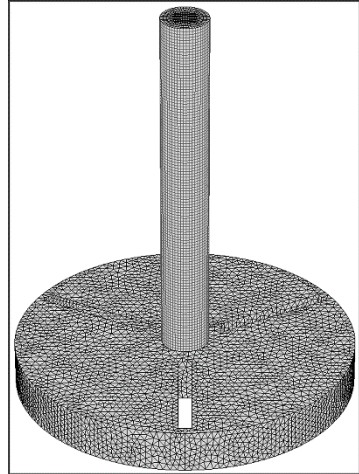

**Figure 6.** Computational domain mesh.

To obtain the number of grid cells required to ensure calculation accuracy and not waste computing resources, five sets of grids were drawn for verification and selection. The effect of the number of mesh nodes between 1,177,438 and 1,997,585 was investigated. Grid-independence analysis was performed by comparing the wear rates of the same blade surface. The particle mass concentration was 1%, the inlet speed was 9.03 m/s, and the disk speed was 400 rpm. The calculation result is shown in Figure 7. When the number of mesh nodes was above 1,329,922, the wear rate reached a stable value, and thus, 1,329,922 nodes were selected for the simulations.

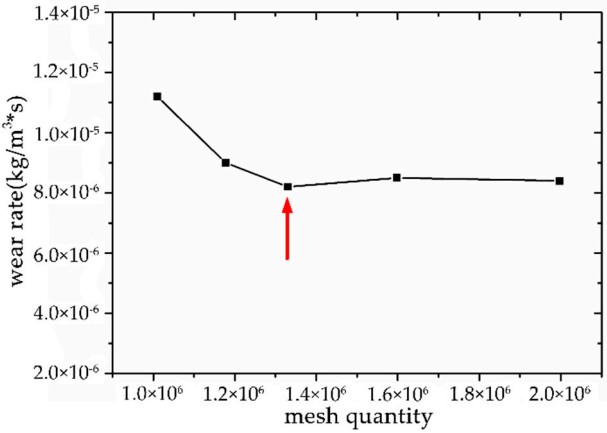

**Figure 7.** Mesh independence test.

### 3.4. Calculation Method Verification

To verify the accuracy of the simulation results, the numerical and experimental results were compared for a disc rotation speed of 500 rpm and a particle mass concentration of 10%. The wear rate of simulation is the mass lost per unit time on a single surface area.

The wear pattern obtained by the experiment is shown in Figure 8. The wear area of the blade working face was divided into the wear area at the blade outlet and the unworn area at the blade inlet. The numerical calculation results of the wear are shown in the upper part of Figure 8. The qualitative analysis showed that the simulation results were in good agreement with the experimental blade wear. The simulated and experimental wear areas were the same. In the simulation results, the blade exit area was significantly worn, corresponding to a high-wear-rate area. This was consistent with the high-wear-rate area of the aluminum sheet in the experiment. In the simulation, the low-wear-rate area of the blade inlet was also consistent with the results obtained from the experimental aluminum sheet.

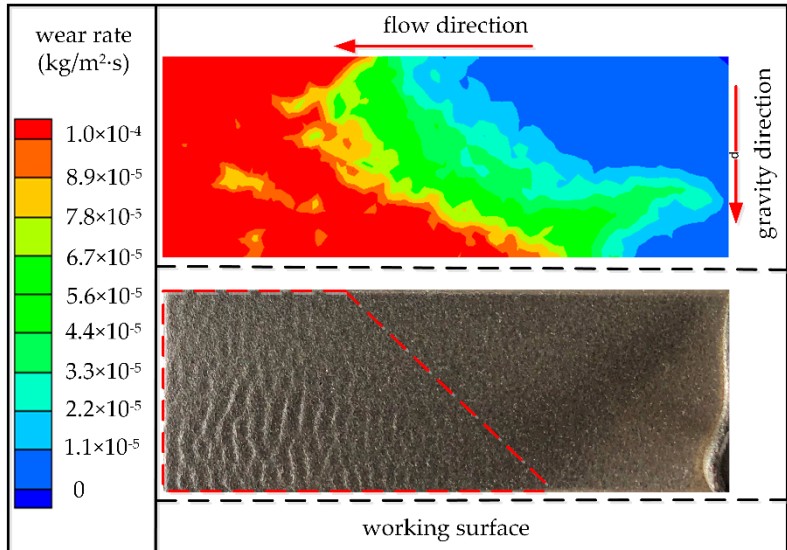

**Figure 8.** Comparison of wear area of blade working face at 500 rpm and 10% concentration.

To further analyze the wear pattern on the blade quantitatively, the thickness loss rates on the three measuring lines of the same blade working face under the above working conditions were extracted for comparison, as shown in Figure 9. Overall, the trends in the simulation results on each measurement line and the experimental results were consistent. The thickness loss rate from the blade inlet to the blade outlet gradually increased. The acceleration of particles through the disc increased the particle velocity in the outlet area of the blade working face, and the collision velocity between solid particles and the outlet area increased, resulting in significant wear in the outlet area.

In addition, the experimental thickness loss rate at the blade exit was smaller than of the simulation. The reason was that the effect of particle breakage was not considered in the numerical simulation process, and the particle movement process during the experiment was more complicated. It is necessary to account for factors such as particle breakage and surface damage due to the particles. Thus, it was concluded that the experimental thickness loss rate was lower than the simulated value. The thickness loss rate extracted experimentally at the exit tended to $2 \times 10^{-5}$ mm·s$^{-1}$.

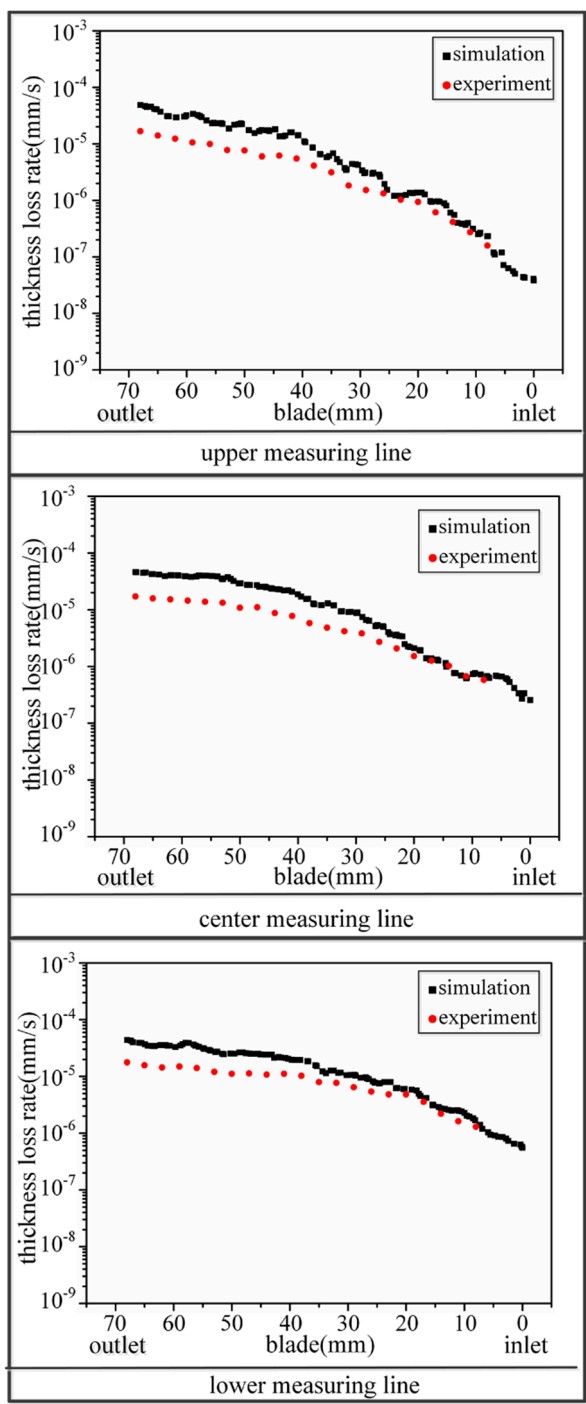

**Figure 9.** Thickness loss rate of No.1 blade working surface at 500 rpm and 10% concentration.

## 4. Results and Analysis

### 4.1. Effect of Rotation Speed on Particle Distribution and Volume Concentration

A Lagrangian coordinate system was used to track the trajectories of the particles. The upper side of Figure 10 shows the particle distribution diagram of different rotation speeds at an 8% particle mass concentration. The lower side is the corresponding particle volume fraction graph under the same working conditions.

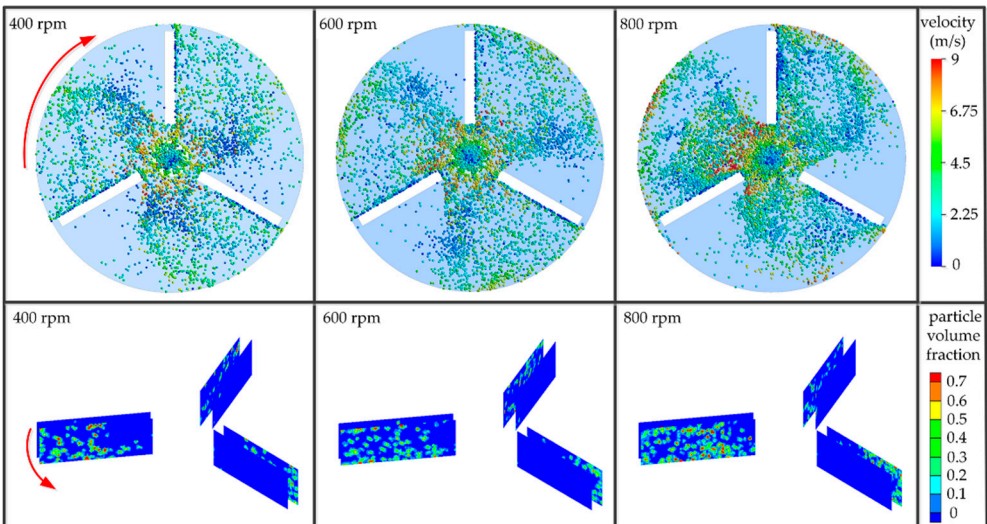

**Figure 10.** Particle distribution and particle volume fraction at different rotation speeds at 8% particle mass concentration.

In the particle velocity distribution, a clockwise rotation was evident from the bottom to the top. As the rotation speed increased, the particle distribution near the blade working surface increased, and the overall particle distribution in the flow channel shifted toward the blade working surface. The particle velocities were significantly reduced near the blade working surface, because the particles had an initial velocity when entering the rotating disk flow path. The collision at the blade inlet caused the particle speed near the blade working surface to decrease. The collision blade was a straight blade, and the collision angle between the particle and the working surface was large. After the particle collision, the particle movement direction changed, and the particle speed decreased at the working face.

When the rotation speed was 400 rpm, the particle velocity near the front edge of the blade increased. The particle velocity in the flow channel away from the blade working surface was relatively small. The particle velocity at the outlet of the flow channel increased, and the velocities were mainly in the range of 2–6 m/s. When the rotation speed was increased to 800 rpm, the particle velocities in the flow channel increased overall. The particle velocity at the outlet of the flow channel increased to the range of 4–9 m/s. The higher the rotation speed of the disk, the higher the kinetic energy provided by the blade wall to the particles. The overall particle velocity in the flow channel increased, especially at the outlet of the flow channel.

When studying the effect of the speed on the particle volume fraction, a counterclockwise rotation was observed from top to bottom. At low speeds, the particles were sparsely distributed on the blade working surface. The particle distribution at the blade outlet was wider than that at the blade inlet. This indicated that particles accumulated at the blade outlet on the working surface at low speeds, and the probability of collision between the particles and the outlet was greater. As the rotation speed increased, the particle distribution area increased and extended toward the blade inlet. The particle distribution at the blade inlet gradually increased. Thus, the probability of particle collision with the working surface increased with the increase in the rotation speed, especially at the blade inlet. However, the volume fraction value of the solid particles did not change significantly with the increase in the rotation speed. This indicated that the increase in the rotation speed only expanded the particle volume distribution range on the working face of the blade and had no influence on the volume fraction of the solid particles.

However, the particles did not collide with the non-working surface of the blade during the entire operation. The volume fraction of solid particles on the non-working surface of the blade was almost zero as the speed increased. The main reason for this was that the flow passage of the rotating disk

expanded greatly, and the kinetic energy of the particles in the flow passage decreased continuously after the collision between the solid particles and the working face of the blade. The particles did not reach the non-working face and collided with it.

Figure 11 shows the wear cloud map with an 8% particle mass concentration at 600 rpm. The wear at the exit of the working face was more significant, and the lower wear range was larger. Wear was not evident on the non-working face. Based on the particle velocity distribution under the same operating conditions and particle volume fraction on the blade, the particle velocity increased near the leading edge of the blade's working face, and the relative velocities between the particles and the wall were relatively large. Thus, the wear rate reached the maximum at the blade outlet edge. Based on the distribution of the particle volume fraction, under the action of centrifugal force and gravity, the particles at the outlet were more distributed than those at the inlet. The number of particles at the lower part of the blade was greater, resulting in a larger wear area at the outlet of the blade face. The lower wear area was larger than the upper wear area, and the overall wear area had a trapezoidal shape. Because the non-working surface did not collide with the particles and the particle volume was almost zero, the non-working surface exhibited no evident wear phenomenon.

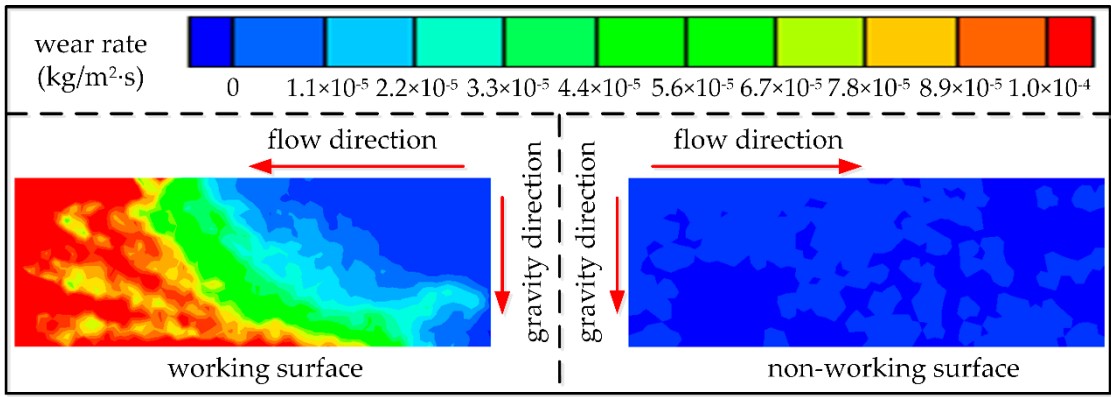

**Figure 11.** Wear cloud diagram at 600 rpm at 8% particle mass concentration.

### 4.2. Effect of Rotation Speed on Blade Wear

Figure 12 shows the wear data of the blades at the same position under different rotation speeds with an 8% mass concentration. The wear zone developed closer to the blade inlet as the speed increased. Under various operating conditions, the thickness loss of the experimental aluminum sheet gradually increased from the inlet to the outlet. The difference in thickness loss at various positions of the blade increased as the speed increased. When the speed was 400 rpm, the aluminum sheet on the pressure surface was only worn significantly at the outlet, but no evident corrugated wear texture was produced. The thickness loss of the experimental aluminum sheet gradually increased from the inlet to the outlet. When the rotation speed was 600 rpm, the corrugated wear texture formed at the outlet of the aluminum sheet on the pressure surface. A relatively stable thickness loss area began to appear at the outlet of the aluminum sheet. The higher the rotation speed, the larger the thickness loss stable area. Of the three measuring lines, the area of the thickness loss of the lower line was the largest. When the rotation speed was increased to 800 rpm, the corrugated wear texture developed to the blade entrance. This was mainly because the collision position of the particles moved to the entrance as the rotation speed increased. The increase in the collision velocity resulted in an increase in the surface wear, so the wear area increased at high speeds. A relatively stable area of wear appeared at the exit. The reason was that the number of particles increased in the vicinity of the lower wall surface of the flow channel after the rotation speed increased, and a particle protective layer formed to reduce the degree of wear damage. Thus, a stable region of thickness loss formed. The severely worn areas of the experimental aluminum sheet were consistent with the severely worn areas in the simulations at various speeds.

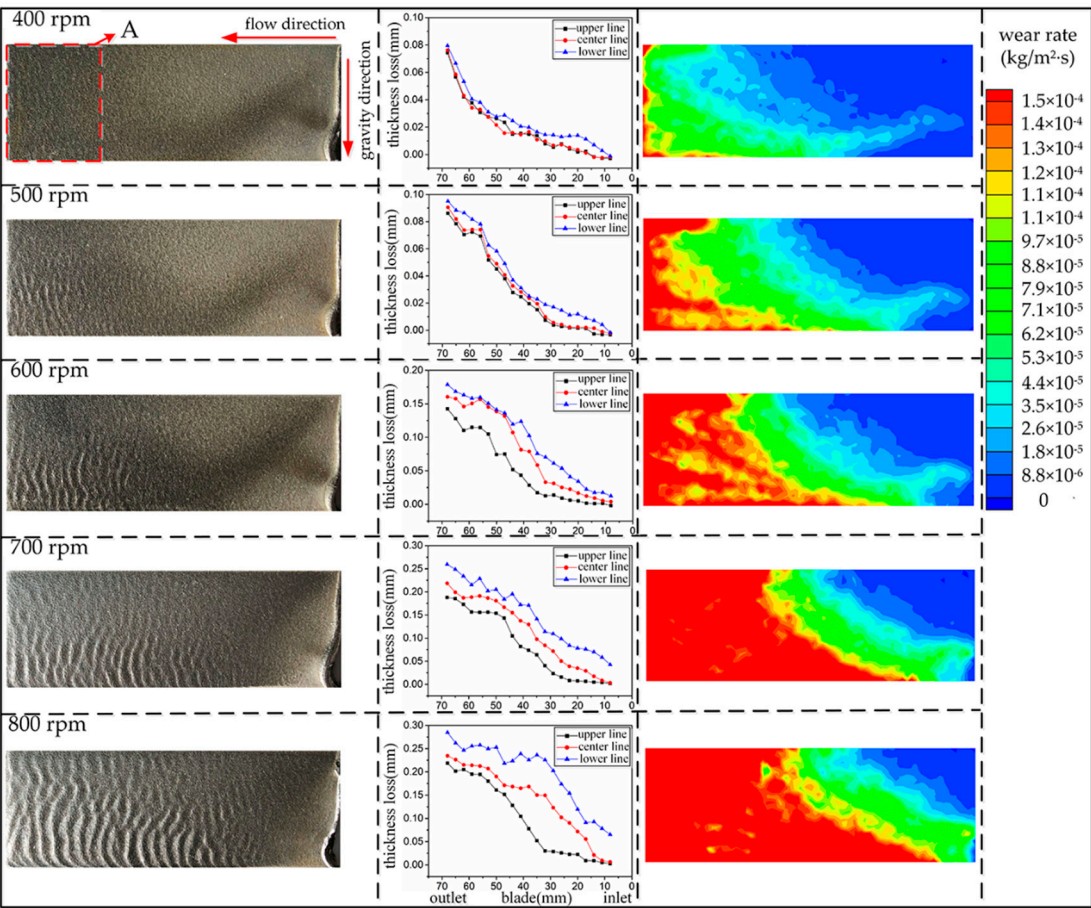

**Figure 12.** Abrasion diagram of the working surfaces of blades at different rotation speeds and an 8% particle mass concentration.

Area A of the working surface of the aluminum sheet shown in Figure 12 was severely abraded. Figure 13 shows the variations in the average and maximum thickness loss rates along the three measurement lines in area A with the rotation speed. The maximum and average thickness loss rates increased with the rotation speed. The growth trends were the same. The maximum thickness loss rate under each working condition was 1.1–1.7 times the average wear rate. It was further confirmed that, as the rotation speed increased, the relative speeds of the particles and the blade wall surface increased, resulting in an increased blade thickness loss rate.

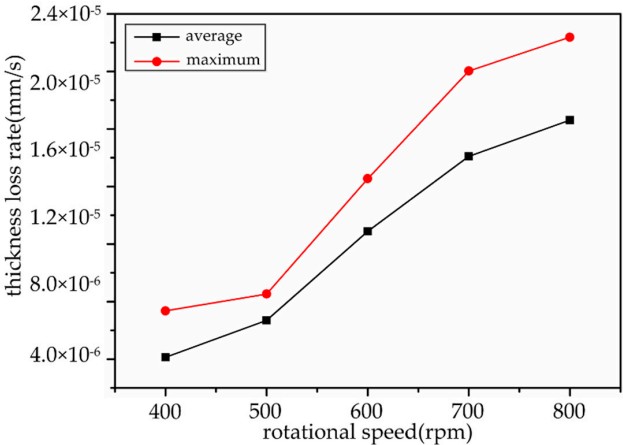

**Figure 13.** Thickness loss rate at the exit area A of the blade working face at an 8% mass concentration.

### 4.3. Effect of Particle Concentration on Blade Wear

Figure 14 shows the wear data and particle volume fractions of the blades at the same position with different mass concentrations at 600 rpm. The increase in particle mass concentration caused increased wear at the exit of the aluminum sheet on the working face, and the wear area developed near the entrance of the aluminum sheet on the working face. At an 8% mass concentration, wear ripples formed at the exit area of the aluminum sheet on the working face. However, compared with the corrugated wear area formed at the exit area at a 10% mass concentration, there was no significant change. Based on the measured thickness loss of the aluminum sheet on the working face, the thickness loss of each measurement line of the aluminum sheet increased with the increase in the mass concentration. The measurement of the lower line at the same position had the largest thickness loss compared to the upper and center lines. However, at 8% and 10% concentrations, the thickness losses on each measurement line were relatively close. In addition, a stable area of thickness loss also appeared along the middle and lower lines of the measurement. At the same rotation speed, the particle distribution of the blade working surface at mass concentrations of 2–10% extended from the blade inlet to the outlet. This showed that, with the increase in the particle inlet mass concentration, the distribution area of the particles was approximately the same. However, the particle volume fraction of the blade working surface gradually increased for mass concentrations from 2% to 10%. As the concentration of solid particles increased, the particles gathered more on the blade working surface, and the frequency of collision between the particles and the wall surface increased. The particle volume fraction from the inlet to the outlet of the blade working face gradually increased, and the particle volume fraction from the middle to the outlet of the blade face was significantly greater than that at the inlet. This indicated that the particles mainly gathered from the middle of the blade to the outlet on the blade working surface, which caused significant wear in this area.

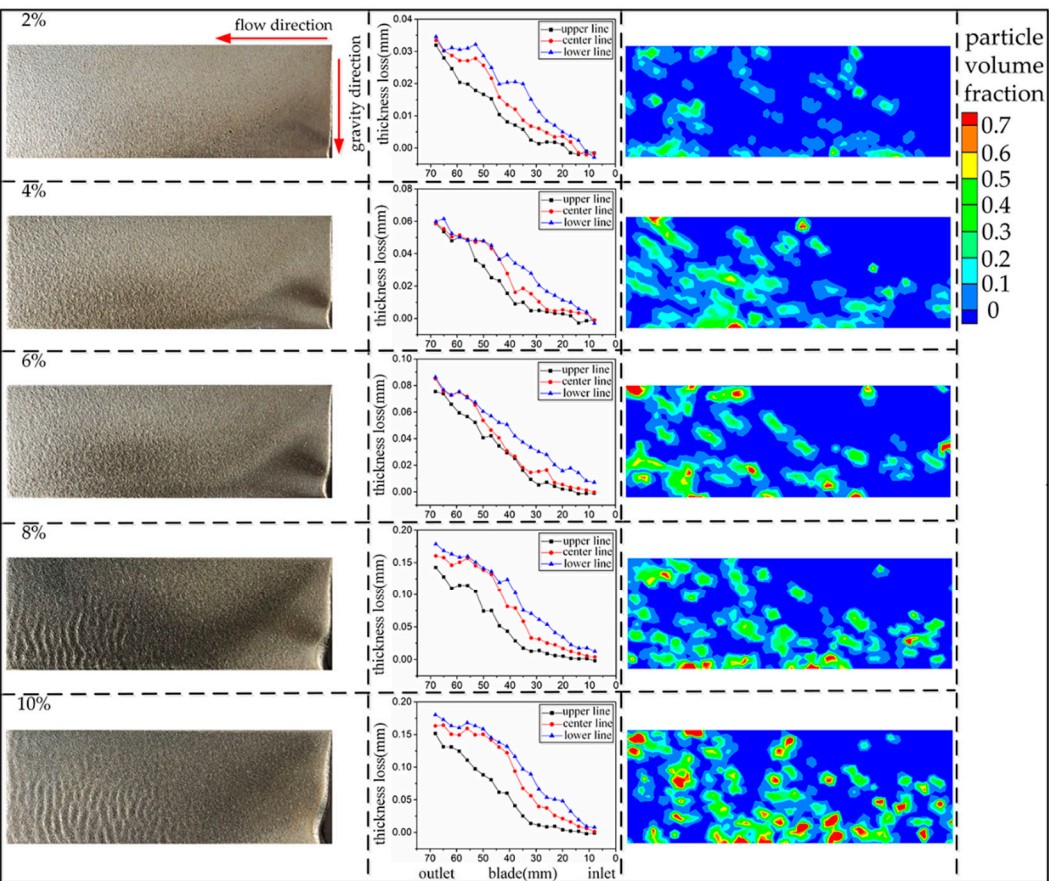

**Figure 14.** Wear for different particle concentrations at 600 rpm.

As shown in Figure 15, at 600 rpm, the maximum and average thickness loss rates of the aluminum sheet outlet area A of the working face increased with the increase in the mass concentration. When the concentration increased to 8% and 10%, the maximum and the average thickness loss rates tended to stabilize. When the particle concentration increased to more than 8%, the particles near the outlet of the working face accumulated to form a buffer layer, and the thickness loss rate at the outlet tended to be stable. The maximum thickness loss rate was 1.1–1.3 times the average thickness loss rate under each working condition.

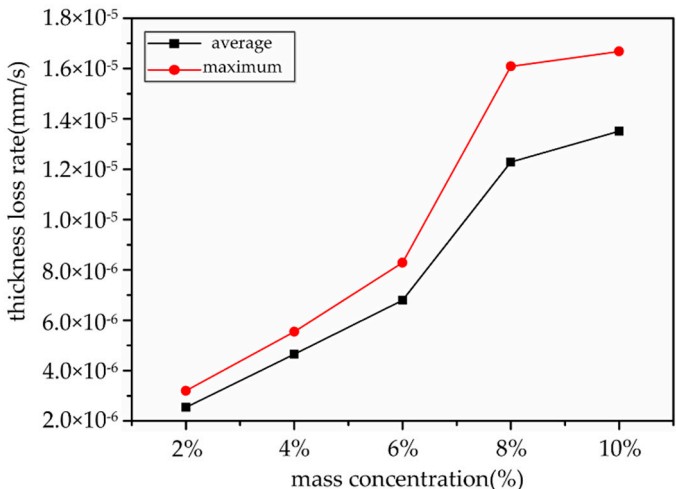

**Figure 15.** Thickness loss rate at the outlet area A of the blade working surface at 600 rpm.

*4.4. Analysis of Different Micro-Topographies of Wear Surface Caused by Different Rotation Speeds*

Figure 16 shows the micro-topography of the lower area of the aluminum sheet at the exit of the working face under different working conditions. As the disk speed increased, the surface of the aluminum sheet gradually changed from a pit shape to a corrugated shape. The amplitudes and spacing of the ripples were not different under each set of conditions. The wear process can be explained using the micro-cutting theory. The aluminum sheet was a soft metal. The solid particles collided with the wall surface. The wall surface plastically deformed to form pits. During the continuous collision, the pits became deeper, and protrusions occurred around the pits. The protrusions peeled from the surface under the impact of the particles.

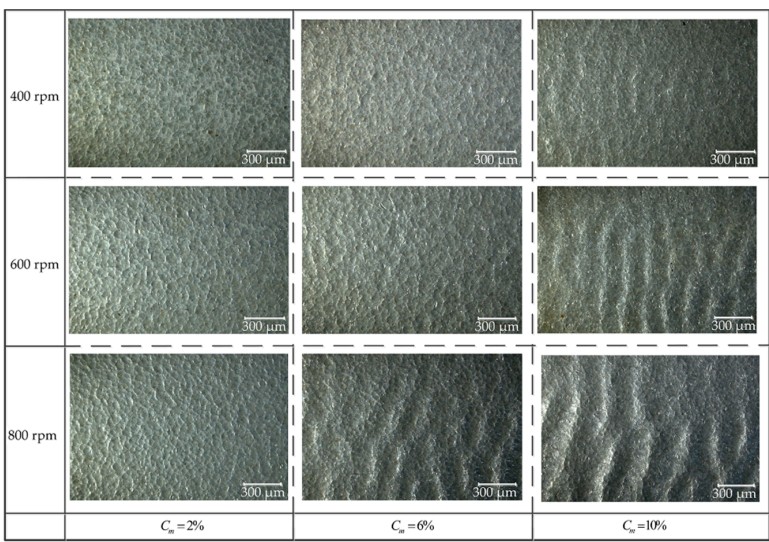

**Figure 16.** Blade wear surface morphology under different working conditions.

Comparing the wear ripples at particle concentrations of 6% and 10% and a rotation speed of 800 rpm, the total width of the five waves was measured. The widths of the five ripples at 6% and 10% were 5.86 and 7.83 mm, respectively. The greater the mass concentration of the particles was at the same speed, the greater the distance between the ripples became. At a 10% mass concentration, the five corrugation widths for rotation speeds of 600 and 800 rpm were 5.46 and 7.83 mm. The corrugation distance of the worn aluminum sheet became larger as the disk speed increased.

Figure 17 shows the wear morphologies of the exit area of the aluminum sheet on the working face at each rotation speed with a 10% mass concentration. At 400 and 600 rpm, the area of the gullies produced by the surface wear of the aluminum sheet was smaller than that produced at 800 rpm. Combined with the movement characteristics of the particles, when the disk speed increased, the particles on the blade working surface increased, and the relative speeds of the particles and the blade working surface increased. The depths of the pits produced by single particles impacting the experimental aluminum sheet increased. The greater the number of particles, the heavier the experimental aluminum sheet wear.

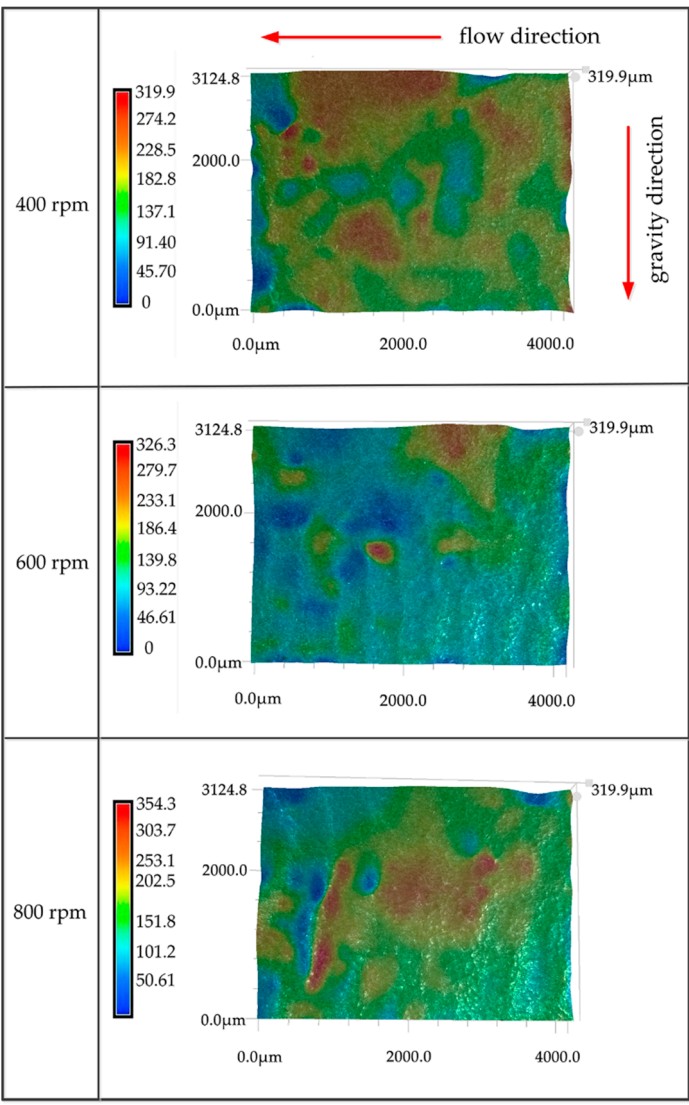

**Figure 17.** Gully heights of aluminum sheet surfaces at different rotation speeds and 10% mass concentration (unit: μm).

With the increase in the disk speed, the gully height difference on the surface of the aluminum sheet increased along the flow direction of the solid–liquid two-phase flow. The increase in the speed



caused the particle velocity at the outlet edge to increase. Deeper pits were generated after a large number of high-speed moving particles collided with the outlet edge continuously. The greater the wear on the surface of the aluminum sheet, the deeper the pits along the flow direction.

## 5. Conclusions

The main findings of this study were as follows.

(1) The kinetic energy of the particles increased with the increase in the disk speed, and the particle speed at the outlet of the blade working surface increased significantly. With the increase in the rotation speed under the same mass concentration, the particle distribution approached the blade working surface, and the accumulation of particles on the blade working surface increased. Due to the expansion of the three-blade flow channel, the non-working surface of the blade rarely collided with the particles.

(2) At low speeds, solid particles accumulated at the outlet of the blade working surface. With the increase in the rotation speed, the particle distribution area increased. The distribution extended toward the inlet of the blade, but the volume fraction of the solid phase particles did not change significantly with the increase in the rotation speed. However, the volume fraction of solid particles increased with increasing particle concentration, and there was no significant change in the wear area.

(3) As the rotation speed increased, the wear area extended toward the blade inlet. However, when the particle concentration was greater than a certain value, the wear area did not change with the speed and tended to be stable. The wear rate of the working face of the blade increased with the increase in the rotation speed, and a stable region of the wear rate appeared at the lower position of the blade outlet. The stable area of the wear rate developed toward the inlet of the blade as the speed increased. There was almost no wear on the non-working surface of the blade.

(4) The aluminum sheet of the working face was significantly worn at exit area A. As the disk speed and particle concentration increased, the wear degree of area A significantly increased, and the growth trends of the maximum and average thickness loss rates in this area were the same.

(5) Increasing the rotation speed of the disk caused the distance between the wear ripples to increase. The aluminum sheet in the exit area of the working face followed the flow direction of the solid–liquid two-phase flow, and the resulting wear grooves gradually deepened, causing significant wear and damage to the exit area of the working face.

**Author Contributions:** Conceptualization, X.Z.; Data curation, P.W.; Formal analysis, P.W.; Investigation, X.Z.; Methodology, P.W.; Project administration, Y.L.; Software, P.W.; Supervision, Y.L.; Writing—original draft, P.W.; Writing—review and editing, X.Z. and Y.L. All authors have read and agreed to the published version of the manuscript.

**Funding:** This work was supported by the National Natural Science Foundation of China (No. 51976197), and Key Research and Development Program of Zhejiang Province (No. 2020C03099).

**Conflicts of Interest:** The authors declare no conflict of interest.

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
