# Peer review of "Analysis of Flow and Wear Characteristics of Solid–Liquid Two-Phase Flow in Rotating Flow Channel"

_processes, doi:10.3390/pr8111512_

Round 1
Reviewer 1 Report
In general:
-English should be improved
-The form of the formulas and equations must be reviewed, as the reference to the variables.
In details:
Introduction:
Major:
-Please emphasize the novelty of the work, i.e. what is the difference between this research and the others in the literature.
- The itemized articles are "categorized" by the particle size, but the disk/impeller size and RPM range is not reported, which I think necessary for the adequate comparison.
- CFD studies missing (see the work of Dr. Gupta, CFD prediction of erosion wear...)
Minor:
I think this should be written in an other way: 0.5-mm-diameter
Experimental setup:
- Error analysis for the laser displacement thickness measurement should be taken. The method is not detailed or referenced.
- In Figure 2 m4 and 5 the labelling should be the same font (and size) as the text.
- Figure 3 has very tiny labelling, polyline is unnecessary
Model design:
I cannot understand: Solidworks was used to model the rotating disk to to simulate the water flow. To create the 3D model of the disk?
How do you checked if the two-phase flow is developed?
Numerical methods:
-“calculated by considering the solid phase as a discrete phase and the fluid phase as a continuous phase in a Eulerian coordinate system.”, however the equations are in Descrates coordinate system. Maybe Eulerian approach?
Mathematical formulation:
-notation must be cleared. i.e. P as eqivalent pressure, The levi-civita symbol etc. You may consider to write the equation in tensorial form firstly. (Eq.:2)
-DPM equations are missing.
-Eq.:4 needs a citation, I think. What is ER?
-Simulation parameters: Why these settings were used?
section 3.3 was good.
-Calculation method verification: Wear rate is not defined.
-blue background is bothering. It is not needed
-figure 9 labelling is tiny.
Results and analysis:
-Figure resolution should be changed, the background and the label front also should reviewed.
Question:
-How the damaged blade influence the flow characteristic and the further wear?
-What do you except if the particle density increasing?
-How the viscosity influences the wear?
Reviewer 2 Report
The paper presents an experimental and modelling study about the flow and wear characteristics of solid-liquid two-phase flow in a rotating flow channel.
Simulation results were compared con experimental data and a detailed summary of results (both experimental and simulated) is presented.
The paper is clearly written and well presented and add some novelty in the field of solid-liquid centrifugal pumps fluid-dynamics. Results can be useful and interesting for industrial people and researchers of the field.
For these reasons, I think that it is worth publishing the article.
Round 2
Reviewer 1 Report
Dear Authors,
Thank you for the answers to my questions! I accept it all.
I recommend a new round of revision, because of the quality of the figures. Please go through on all the figures and equalize the font sizes! Figure 7. and 9. are good examples, but on figure 8., the font size is much bigger, figure 3. (b) is much smaller... The font family should be the same as well.In figure 17. the black background and the size of the color bar's font are also a problem. Please fix it!
The first things that a reader looks at in an article are the figures, that is why it is important.
